# A protocol to develop a Quality Assessment Tool for Realist Synthesis (QUATRES)

Ferdinand C. Mukumbang[1]*, Brynne Gilmore[2,3], Rebecca Hunter[4], Sara Dada[2,3], Sonia Dalkin[5], Andrew Booth[6,7]

**1** Department of Global Health, School of Public Health, University of Washington, Seattle, Washington, United States of America, **2** UCD Centre for Interdisciplinary Research, Education and Innovation in Health Systems (UCD IRIS Centre), Dublin, Ireland, **3** School of Nursing, Midwifery and Health Systems, University College Dublin, Dublin, Ireland, **4** Department of Public Health and Sport Sciences Medical School, University of Exeter, Exeter, England, **5** Faculty of Health and Life Sciences, Northumbria University, Newcastle Upon Tyne, United Kingdom, **6** Sheffield Centre of Health and Related Research (SCHARR), University of Sheffield, Sheffield, United Kingdom, **7** University of Limerick, Limerick, Ireland

\* ferdie@uw.edu

## Abstract

A realist synthesis or review is a theory-driven, realist-informed interpretive approach to synthesizing secondary data, informing evidence-based practice, and explaining social phenomena. Realist syntheses use flexible and iterative methods to achieve this goal, including drawing on stakeholders' knowledge. Thus, realist syntheses require robust planning to be conducted and reported. Similarly, owing to their complex and iterative nature, they also need assessment guidelines so that knowledge practitioners and policymakers can appraise the value of the evidence they produce. While some current tools guide the conduct and reporting of realist syntheses, little comprehensive guidance exists on effectively assessing the quality of realist syntheses. To this end, we aim to develop a Quality Assessment Tool for Realist Synthesis (QUATRES) to encourage the transparent reporting of the review methods, the explicit assessment of the quality of reviews and the reliability of their findings. To achieve this, we will employ a multi-method study design consisting of three work packages: an audit of peer-reviewed articles to identify strengths and weaknesses in a sample of papers reporting realist syntheses, a methodological review of realist synthesis methodology guidance to identify quality markers, and a Delphi consensus to refine and validate quality markers.

## Background

A realist synthesis or review is a theory-driven interpretive approach to synthesizing secondary empirical data to inform evidence-based practice and explain social phenomena [1–4]. First proposed by Pawson in 2002 [5], realist syntheses are increasingly used to understand how, why, for whom, and under what conditions policies and

**Data availability statement:** This is a study protocol to design an assessment tool for realist reviews. The findings from the delphi processes will be published alongside the full papers.

**Funding:** This study was supported by the Health Research Board (Ireland) and the HSC Public Health Agency (Grant number ESI-2021-001) through Evidence Synthesis Ireland/ Cochrane Ireland.

**Competing interests:** The authors have declared that no competing interests exist.

programs work. This understanding is achieved by developing program theories that specify the underlying assumptions about how a program or intervention is supposed to work. A key strength of realist synthesis is theorizing the underlying mechanisms and causal pathways through which change occurs when policies or programs are introduced in specific contexts. By providing theory-informed recommendations, realist syntheses offer a deeper understanding of how and why programs produce outcomes, addressing some of the limitations of conventional review methods [6]. As a result, realist syntheses are increasingly used to improve our understanding of how programs work or why they fail in different contexts.

The interpretive nature of realist syntheses is grounded in the realist philosophy of science. While realist syntheses, like other systematic reviews, require methodological rigor and practical relevance, they also prioritize philosophical engagement [7]. Four important principles of realist methodologies relevant to research and reviews include stratified ontology, mechanism-based causality, emergence, and open systems [8]. Critics of realist syntheses have argued that their philosophical approach, while insightful, can be overly theoretical and of little practical consideration [7]. A flexible and iterative approach that engages key stakeholders throughout the research process is necessary to reconcile the theoretical depth of realist syntheses with the practical needs of policymakers, practitioners, and patients. To this end, realist researchers resist using overly prescriptive approaches while conducting realist syntheses [1]. The iterative and flexible nature of conducting realist syntheses while integrating contributions from various stakeholders poses unique challenges for planning and conducting realist syntheses compared to other evidence synthesis approaches.

Reporting standards and quality assessment (critical appraisal) guidance are equally important instruments to guide researchers and practitioners involved in research, evidence synthesis, and policymaking [9]. While reporting standards provide researchers with a list of items to consider while conducting and reporting realist syntheses, critical appraisal tools allow evidence users (researchers, practitioners, and funders/commissioners of research) to evaluate the evidence obtained from realist syntheses [9,10]. A quality assessment tool encapsulates a careful and systematic process for evaluating a study's trustworthiness and methodological rigor. It helps to determine whether an individual can have confidence in study findings and their trustworthiness, value, and relevance for a particular purpose [11,12]. Quality assessment of reviews and other forms of evidence synthesis has become a mainstay when ascertaining the quality of the output, which can be used to inform policymaking and practice. Consequently, established review and synthesis approaches, such as systematic reviews, narrative reviews, scoping reviews, and qualitative evidence synthesis, have purpose-specific critical appraisal tools that consistently assess methodological robustness [13–16]. For instance, quality standards for meta-synthesis research include conducting a comprehensive literature search, rigorously assessing the quality of included studies, using a transparent and systematic data extraction process, analyzing data with a well-defined theoretical framework, and providing a detailed and critical interpretation of the synthesized findings. All the above individual

processes are undertaken to ensure methodological transparency. Thus, quality assessment tools are used to evaluate the validity of the data, completeness of reporting, methods and procedures, conclusions, and compliance with ethical standards, thereby ascertaining the trustworthiness and usability of the evidence.

When Pawson proposed the realist synthesis approach in 2002, he suggested that it offered better usefulness over systematic and narrative reviews but wondered how it would be received by the practice community [5]. He also said, "We will have to wait and see". We are confident that he would be pleased to see that the number of published realist syntheses in the health-related field has increased significantly (2013: 42; 2018: 104; 2023: 194). While realist synthesis was first proposed in 2002, it was not until 2013 that Wong et al. [17] developed the Realist and Meta-Narrative Evidence Synthesis: Evolving Standards (RAMESES) quality and publication standards to guide their robust reporting. The RAMESES project later developed quality standards to assess various aspects of realist syntheses for researchers, peer-reviewers, and funders/commissioners of research, using a four-point scale ranging from "Inadequate" to "Excellent" [18]. A decade after the development of the RAMESES project checklist for appraising realist syntheses, Dada et al. [6] identified challenges in conceptualizing relevance, richness, and rigor when selecting and appraising evidence in realist syntheses. They offered pragmatic suggestions for how realist reviewers can better implement these concepts in practice. Duddy and Wong [2] provided further potential solutions for conducting realist syntheses, explaining relevant concepts and troubleshooting common challenges. Fig 1 illustrates current resources for conducting, appraising, and reporting realist syntheses.

A careful examination of the guidance presented in Fig 1 reveals that it predominantly targets knowledge producers (researchers and funders). What is conspicuously missing are quality assessment tools specifically designed for those who seek to use (i.e., read and apply) the realist synthesis product (output). Even the RAMESES Project, designed to provide quality standards for realist syntheses for researchers, peer-reviewers, and funders/commissioners of research, reveals known inadequacies in its capacity to allow users to assess the quality of realist syntheses. Although realist researchers are reluctant to endorse prescriptive steps to conduct realist syntheses, Tod et al. [12] propose that a robust

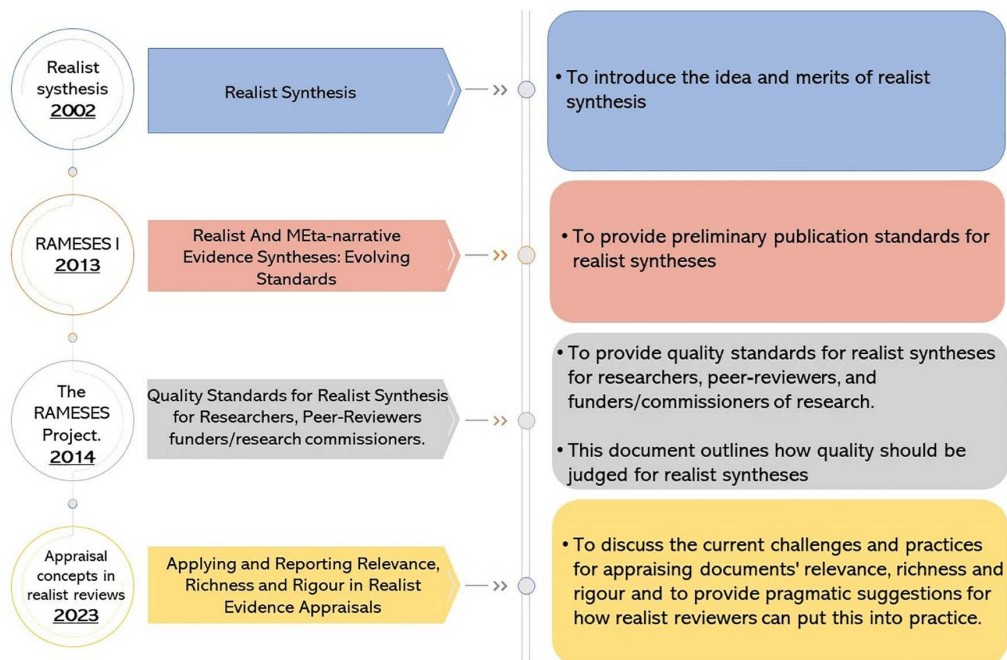

**Fig 1. An illustration of an existing tool for conducting and reporting realist syntheses.**

assessment tool should allow the users of the outputs of realist syntheses to (a) identify the study type(s) of individual evidence included, (b) perform the appraisal, and (c) summarize and use the results. We argue that guiding principles are needed to help users assess the quality and reliability of realist syntheses and their findings. This protocol outlines a plan to develop a comprehensive quality assessment tool for realist syntheses, informing subsequent uptake of the evidence they generate [19].

## Methodology and methods

To develop QUATRES, we plan to adopt the steps proposed by Moher et al. (2010) for developing a health research reporting guideline: (1) initial steps, (2) pre-meeting activities, (3) face-to-face consensus meeting, (4) post-meeting activities, and (5) post-publication activities. While these steps target reporting guidance, not quality assessment tools, Whiting and colleagues [20] convincingly affirm that a similarly constructed and rigorous process should be followed in developing quality assessment tools. These five main steps include sub-activities to be considered and supplemented as relevant for developing QUATRES. These sub-activities are discussed under each of the main steps.

To develop a comprehensive quality assessment tool for realist syntheses, we will adopt a multi-methods study design consisting of three work packages (Fig 2): (1) An audit of a sample of peer-reviewed practical applications of realist synthesis publications to identify strengths and weaknesses in planning and conducting realist reviews; (2) A methodological review of methodological literature on realist syntheses to identify quality markers for conducting and reporting realist reviews; and (3) A three-round Delphi consensus study. We plan for this project to last for a year, and currently, we are conducting a literature search and screening for auditing realist review publications.

### Initial steps

According to Moher et al. [11], the initial steps for developing consensual evidence-based guidance focus on the executive or working group's early activities. An important initial step was forming a working group to develop QUATRES.

1. **Forming a working group.** While often taken for granted, the thoughtful assembly of a working group constitutes a crucial first step. We aimed to form a team with extensive experience and a history of healthy collaboration in realist research, including realist syntheses projects. Five team members (FCM, SoD, RH, SaD, and BG) had previously worked on guidance and recommendations to address the common challenges when assessing the relevance, richness, and rigor of documents included in realist syntheses [6]. These co-authors have extensive experience conducting realist inquiries (realist research, realist evaluations and realist syntheses) and have helped open the black box of conducting realist-informed data collection and analysis [21,22]. The final member of our team (AB) brings extensive experience in developing research tools and instruments, including quality assessment tools [15] and reporting standards [15], and in auditing the reporting of realist reviews, specifically of search methods [23]. Three group members (SaD, AB, and BG) recently mapped the current practice of advisory groups in realist syntheses to guide the planning and reporting of group involvement [24]. Forming the QUATRES working group based on these collaborations was relatively straightforward. As members of this international six-person group and authors of this protocol, we communicate via teleconference, e-mail, and face-to-face meetings. We have also engaged with members of the original RAMESES Project who are involved in developing the supporting materials for the RAMESES quality standards for realist synthesis and will continue to do so throughout the study [18,25].

2. **Establishing the need for a guideline.** Moher et al. [11] advocate that the first responsibility of the working group is to clearly and explicitly set out their objectives and scope of recommendations for the quality assessment tool. To achieve this, we propose the following activities: (1) to compare the process and the content of a critical review for systematic reviews with meta-analysis [A Measurement Tool to Assess Systematic Reviews—AMSTAR] [16] with the current standard for reporting realist synthesis (RAMESES quality and publication standards) developed by Wong et al. [17]. Our goal is to compare the content in the AMSTAR critical appraisal tool with that in the RAMESES quality and

publication standards to identify any missing elements that need to be addressed in a critical appraisal tool for assessing realist syntheses.

Moher et al. [11] recommend searching for relevant evidence on the quality of reporting in published research articles. Therefore, we plan to audit a sample of peer-reviewed realist syntheses to identify areas for improvement in their

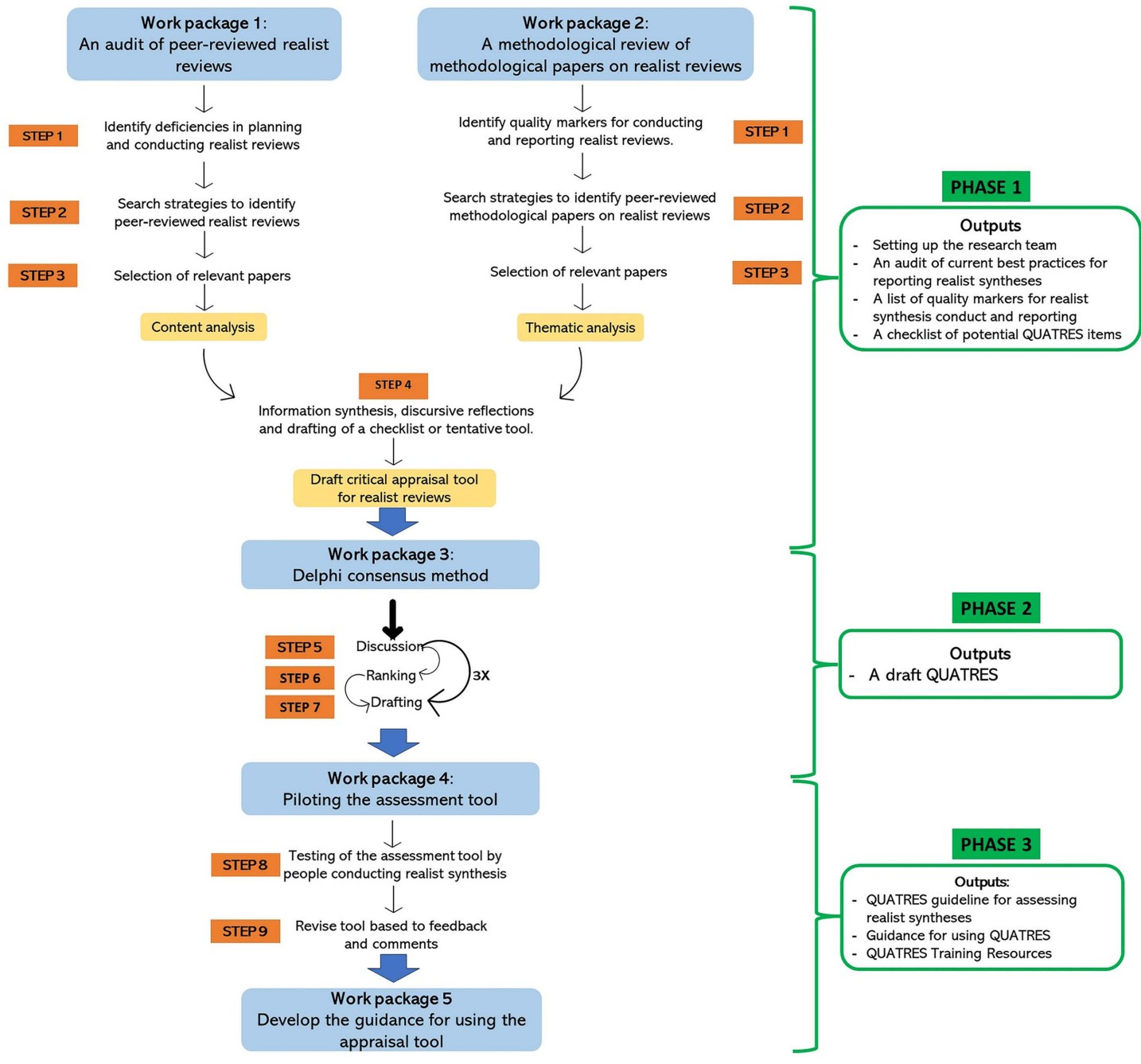

**Fig 2. Proposed steps for developing a quality assessment tool for realist synthesis.**

planning, conducting, and reporting. The output of this audit will be a narrative review. We plan to conduct a snapshot of realist syntheses published in 2024 to examine current practice for realist syntheses and assess how the appraisal, analysis, and synthesis of the selected studies were undertaken and reported. Our assessment will focus on the fidelity with which the authors of the sampled realist syntheses report components of their realist synthesis against the reporting standards stipulated in RAMESES I quality and publication standards.

Identifying key information about the potential sources of bias in relevant studies [11] is also critical for establishing the need for QUATRES. Quality markers are standardized, evidence-based measures to be used to assess the quality of research. To this end, we plan to systematically review methodological papers on realist syntheses to identify quality markers for realist syntheses. This methodological review will address how to judge the quality of realist synthesis. Identifying quality markers will involve conducting a comprehensive methodological review of methodological papers and methods-rich peer-reviewed health- and social care-related realist syntheses within the last ten years. The output of this stage will be a list of realist quality markers, including identifying any gaps that require further development.

**3. Obtain funding for the guideline initiative.** Obtaining funding to generate quality reporting standards is one of Moher and colleagues' recommendations to those embarking on this journey. For the early phase of developing QUATRES, we obtained the Evidence Synthesis Ireland (ESI) Study within a Review (SWAR) seed funding from the Health Research Board (Ireland) and the Health and Social Care Public Health Agency (Northern Ireland). This fund will enable us to employ a research assistant to provide overall research support for the audit and methodological reviews proposed in Work Packages 1 and 2. The fund will also support the travel expenses for team members to meet in person when resolving conflicts, extracting and collating quality markers, and developing draft markers to inform the development of a Delphi survey. Funds will be used to print relevant knowledge translation resources within facilitated workshops or conferences. As informed by the outputs from this first phase, we plan to apply for a larger grant to facilitate the project's second phase.

### Pre-meeting activities

Outcomes of Phase I will include established processes for the QUATRES team to work together, an audit of current best practices for reporting realist syntheses, and a list of quality markers for conducting and reporting realist syntheses. While the outputs produced in phase I undergo further testing and refinement, they can be used as interim resources by the realist, policy, and practice communities while Phase II is completed. Phase II will utilize the resources developed in Phase I to refine, review, and pilot QUATRES. We will conduct a three-round Delphi study to gather feedback and refine the tool.

**4. Discuss the rationale for including items in the checklist.** The research team plans to meet face-to-face to establish items for inclusion in the quality assessment tool. According to Moher et al. [11], the guideline should include checklist items. During the development of quality standards and methodological guidance for Realist and meta-narrative reviews, Greenhalgh et al. [22] conducted a literature review to inform the development of their checklist, which served as the basis for their online Delphi discussions. They performed a comprehensive search to identify real examples of realist review activities. Through this process, they could scrutinize and formalize these activities into a checklist for conducting and reporting realist syntheses. Similarly, during our anticipated face-to-face meeting, we will integrate the findings of the 2024 audit of realist syntheses with the methodological review of quality markers and develop a checklist to inform the project's next phase.

**5. Identifying Participants.** An international multidisciplinary group is required to develop guidelines and appraisal tools. The expertise of the multidisciplinary group's individuals should align with the specific guidance being considered [11]. While developing the RAMESES publication standards for realist syntheses, Wong et al. [17] conducted a Delphi study with 33 members. These 33 members comprised researchers in public or population health research, evidence synthesis, health services research, international development, education, research methodologists, publishing, nursing,

and policy and decision-making. For QUATRES, we plan to conduct online Delphi surveys with individuals with similar backgrounds to those who participated in the RAMESES publication standards for realist syntheses. We plan to identify and enroll these individuals through the existing RAMESES listserv with an estimated 300 subscribers, as well as through other realist communities such as 'The Realist Hub,' 'NoRTH: Northern Realist Research Team Hub,' 'PhD Realist Network,' and others as identified through our networks.

**6. Conduct a Delphi exercise.** Our next step will be to organize a Delphi consensus method—a systematic forecasting process that utilizes the collective opinions of panel members [26]. The Delphi consensus method involves conducting multiple rounds of surveys to facilitate iterative discussions among panel members. While there is no standard size for panel members, it can vary from 10 to 1000, with numbers close to 30–50 considered optimal [26]. We aim to recruit 30–50 participants for the Delphi consensus study. Using the list of quality markers for the conduct and reporting of realist syntheses developed in Phase 1, we plan for three rounds of Delphi consensus, with each selected panel member asked to rate each potential item for relevance and clarity.

### The online consensus meeting

**7. Present and discuss results of pre-meeting activities and relevant evidence.** Our first task during the Delphi meeting will be to review the objectives and outline the meeting's structure, as well as clarify any outstanding issues among the panelists. During the Delphi meeting, we will present the background topics, the rationale for developing QUATRES, the empirical evidence from the audit of current best practices for reporting realist syntheses, and the current limitations, as well as the list of quality markers for conducting and reporting realist syntheses. After each survey round, the data will be analyzed and presented to all panel experts in a clear and understandable format [26]. This analysis will include simple charts and statistics to demonstrate the stability of responses. An 80% consensus threshold will be used to determine the degree of agreement among participants, indicating a shared understanding of the essential components of QUATRES [27]. Each round of the Delphi process will retain components that meet this threshold. By analyzing successive rounds, we can assess data to identify consensus and stability. This iterative approach effectively collects qualitative information, refining statements for panel members and achieving consensus.

### Post-meeting activities

**8. Develop the guidance statement.** The views expressed in a Delphi exercise will be translated into reporting guidelines. Of course, developing a refined reporting guideline requires several iterations. Therefore, after the Delphi exercise, we will adjust the originally drafted checklist after identifying which parts will be retained and which items will be modified or added. The iterative process will aim to craft each item into a crisply and unambiguously worded checklist item [11] to form a first draft of the QUATRES tool. An efficient way to complete such a process is to invite people working on realist syntheses to try out the tool [20]. Once again, we will invite potential reviewers through the RAMESES listserv and other realist communities such as 'The Realist Hub,' 'NoRTH: Northern Realist Research Team Hub,' and 'PhD Realist Network' to try out the tool on realist syntheses on which they are working. For a new reporting guideline, we should produce a concise document outlining the rationale for developing the guidance, the development process, a brief description of the meeting and participants involved, and a checklist. Our report document will be a peer-reviewed journal article published in a peer-reviewed journal.

### Post-publication activities

**9. Dealing with feedback and criticism.** Developing quality standards entails obtaining feedback and criticism from all stakeholders regarding the reporting guidelines. During the development of QUATRES, we will solicit feedback on the developing guideline through various avenues, including realist working groups such as the RAMESES listserv, Realist

and evaluation conferences and symposia, and the QUATRES website. Constructive criticism can help improve the reporting guidelines when an update is considered.

**10. Encourage guideline endorsement and adherence to the guideline.** When peer-reviewed journals, authors, and users endorse guidelines, they are likely to adopt them in their practice. We will seek the endorsement of QUATRES by journal editors, which can be sought through engagements with these editors at relevant conferences and other realist communities such as the RAMESES listserv, 'The Realist Hub,' 'NoRTH: Northern Realist Research Team Hub,' Ph.D. Realist Network and others, as identified through our networks.

It is advised that the endorsement of quality reporting guidelines should adopt specific language to avoid vagueness. For example, BioMed Central, the publisher of over 300 open-access peer-reviewed journals in 2023, states, "We recommend authors refer to the EQUATOR network website for further information on the available reporting guidelines for health research, and the MIBBI Portal for prescriptive checklists for reporting biological and biomedical research where applicable" [11]. In line with this practice, we plan to encourage the endorsement of QUATRES by appealing to journal Editors to refer authors to the QUATRES website and use the QUATRES guideline in tandem with RAMESES guidelines for their submitted manuscripts.

To encourage the adoption and adherence to the guidelines, we also plan to obtain funding to support the construction of a website dedicated to the QUATRES. The Website will contain the QUATRES checklist, statement, and ancillary documents. We will strive to develop the Website before the publication of QUATRES so that the web address can be included in the published articles.

## Discussion

In this protocol, we note that quality standards for realist syntheses had previously been developed through the RAMESES project [18]. Over the decade since the RAMESES Project developed these resources in 2014, important strides have been made to improve the planning and conduct of realist syntheses. Methodological clarity has been provided relating to: conducting evidence searches guided by realist principles [23], selecting and appraising evidence sources [6], conducting data analysis [21,22], deciding which causal configurations to use, how, and why [28], program theory development [29], the nature and contributions of middle range theories obtained from realist syntheses [30], involving advisory groups [24], and aligning the outputs of realist syntheses to the goals of policymakers and practitioners [7]. Our goal for developing this quality assessment tool is to capture any methodological clarifications published since the RAMESES project developed the assessment tools in 2014.

The hallmark of realist syntheses is their potential to provide findings that not only tell policymakers or managers whether something works but also offer insight to the policy and practice community with a detailed and highly practical understanding. The findings of realist syntheses aim to provide a deeper understanding of complex social programs and policies, informing the planning and implementation of these programs at national, regional, or local levels [31]. To achieve these goals, users of realist synthesis outputs should be able to assess the process used to generate findings and the trustworthiness of those findings. The explanatory nature of realist synthesis findings based on applying Realist philosophical tenets and methodological rigor makes realist syntheses challenging for researchers and practitioners who are less familiar with them. All these developments make a quality assessment tool necessary for guiding the quality and appraisal of findings from realist synthesis. This comprehensive and updated quality assessment tool will support those interested in evidence synthesis, those working on realist syntheses, funding panels, journal editors/peer reviewers, as well as evidence-based decision-makers.

## Conclusion

The quality assessment of realist reviews is crucial for informing the adoption of evidence derived from them. Due to the philosophical grounding and methodological flexibility of realist synthesis, we argue that a formal and comprehensive

quality assessment is necessary to facilitate the adoption of evidence generated from realist syntheses, using a structured tool. By linking the underlying philosophical tenets to the empirical practices involved in obtaining evidence in realist synthesis, QUATRES offers the potential to facilitate the translation of realist review findings into practice. This proposal outlines plans and steps to develop a quality assessment tool for the realist synthesis, supporting its adoption.

## Author contributions

**Conceptualization:** Ferdinand C. Mukumbang, Brynne Gilmore, Rebecca Hunter, Sara Dada, Sonia Dalkin, Andrew Booth.

**Funding acquisition:** Ferdinand C. Mukumbang, Brynne Gilmore, Rebecca Hunter, Sara Dada, Sonia Dalkin, Andrew Booth.

**Investigation:** Ferdinand C. Mukumbang, Brynne Gilmore, Rebecca Hunter, Sara Dada, Sonia Dalkin, Andrew Booth.

**Methodology:** Ferdinand C. Mukumbang, Brynne Gilmore, Rebecca Hunter, Sara Dada, Sonia Dalkin, Andrew Booth.

**Project administration:** Ferdinand C. Mukumbang.

**Supervision:** Sonia Dalkin, Andrew Booth.

**Validation:** Ferdinand C. Mukumbang.

**Visualization:** Ferdinand C. Mukumbang.

**Writing – original draft:** Ferdinand C. Mukumbang.

**Writing – review & editing:** Ferdinand C. Mukumbang, Brynne Gilmore, Rebecca Hunter, Sara Dada, Sonia Dalkin, Andrew Booth.

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
