## [Decision Letter · Decision Letter 0]

4 Jun 2025

PONE-D-25-04980A protocol to develop A QUality Assessment Tool for REalist Synthesis (QUATRES)PLOS ONE

Dear Dr. Mukumbang,

Thank you for submitting your manuscript to PLOS ONE. After careful consideration, we feel that it has merit but does not fully meet PLOS ONE’s publication criteria as it currently stands. Therefore, we invite you to submit a revised version of the manuscript that addresses the points raised during the review process.

We look forward to receiving your revised manuscript.

Kind regards,

Nishant Premnath Jaiswal, MBBS, PhD

Academic Editor

PLOS ONE

Journal Requirements:

[This study was supported by the Health Research Board (Ireland) and the HSC Public Health Agency (Grant number ESI-2021-001) through Evidence Synthesis Ireland/Cochrane Ireland.].

3. Thank you for stating the following in your manuscript:

[This study was supported by the Health Research Board (Ireland) and the HSC Public Health Agency (Grant number ESI-2021-001) through Evidence Synthesis Ireland/Cochrane Ireland.]

[This study was supported by the Health Research Board (Ireland) and the HSC Public Health Agency (Grant number ESI-2021-001) through Evidence Synthesis Ireland/Cochrane Ireland.]

[NO authors have competing interests].

6. Please provide a complete Data Availability Statement in the submission form, ensuring you include all necessary access information or a reason for why you are unable to make your data freely accessible. If your research concerns only data provided within your submission, please write "All data are in the manuscript and/or supporting information files" as your Data Availability Statement.

Reviewers' comments:

Reviewer's Responses to Questions

**Comments to the Author**

1. Does the manuscript provide a valid rationale for the proposed study, with clearly identified and justified research questions?

Reviewer #1: Yes

2. Is the protocol technically sound and planned in a manner that will lead to a meaningful outcome and allow testing the stated hypotheses?

Reviewer #1: Yes

3. Is the methodology feasible and described in sufficient detail to allow the work to be replicable?

Reviewer #1: Yes

4. Have the authors described where all data underlying the findings will be made available when the study is complete?

Reviewer #1: No

5. Is the manuscript presented in an intelligible fashion and written in standard English?

Reviewer #1: Yes

6. Review Comments to the Author

You may also provide optional suggestions and comments to authors that they might find helpful in planning their study.

Reviewer #1: I really enjoyed reading this protocol for the development of a quality assessment tool for realist synthesis. It is a well written protocol, following a robust approach, and providing a strong rationale for updating previous quality standards.

I have only minor comments which I hope the authors find helpful.

1. Abstract – line 30 and Background line 58 – “A realist synthesis or review is a theory-driven, realist-informed interpretive approach to synthesizing secondary empirical data”

Is the reference to ‘empirical’ data a bit misleading given that realist synthesis or review synthesize all types of data – not just empirical?

2. Line114 “We will have to wait and see – need to close quotation brackets

3. Line 338 and 332 refers to guidelines – will there be multiple guidelines? Or is this a typo?

4. Line 339 – some typos identified here - ‘Ph.D. Realist Network and others are identified……

Should this be - PhD Realist Network and others as identified……

5. Line 340 – should this sentence be the start of a new paragraph? “It is advised that….” And be linked to line 347 as part of the same paragraph? As it currently reads this section seems somewhat disjointed and needs to link more clearly.

For example, it is not clear how this sentence “endorsement of quality reporting guidelines should adopt specific language to avoid vagueness” links to the next bit about BioMed Central etc.

6. Line 349 – Should this be “The Website will contain ….”

7. Conclusion - this section seems rushed which is unfortunate as the conclusion is so important.

Line 384 – this first sentence in the conclusion seems out of place - is this about “Assessment of the quality of included studies? This is a component of the quality appraisal process but not what this protocol is about?

The following sentence seems to lose the focus of the protocol too – it starts off saying quality assessment is required “owing to the philosophical grounding and methodological flexibility of realist synthesis” – clear enough – but how will this “facilitate adoption of evidence generated from realist syntheses using a structured tool”?

The final sentence essentially says the same thing again but is not clear exactly how plans and steps to develop a quality assessment tool for the realist synthesis will support its adoption.

7. PLOS authors have the option to publish the peer review history of their article (what does this mean? ). If published, this will include your full peer review and any attached files.

**Do you want your identity to be public for this peer review?** For information about this choice, including consent withdrawal, please see our Privacy Policy .

Reviewer #1: **Yes: ** Tracey McConnell

---

## [Author Response · Author response to Decision Letter 1]

26 Jun 2025

I have attached a documents indicating how we addressed the reviewers' comments.

---

## [Decision Letter · Decision Letter 1]

28 Aug 2025

A protocol to develop A QUality Assessment Tool for REalist Synthesis (QUATRES)

PONE-D-25-04980R1

Dear Dr. Mukumbang,

We’re pleased to inform you that your manuscript has been judged scientifically suitable for publication and will be formally accepted for publication once it meets all outstanding technical requirements.

Kind regards,

Nishant Premnath Jaiswal, MBBS, PhD

Academic Editor

PLOS ONE

Reviewers' comments:

Reviewer's Responses to Questions

**Comments to the Author**

1. Does the manuscript provide a valid rationale for the proposed study, with clearly identified and justified research questions?

Reviewer #1: Yes

2. Is the protocol technically sound and planned in a manner that will lead to a meaningful outcome and allow testing the stated hypotheses?

Reviewer #1: Yes

3. Is the methodology feasible and described in sufficient detail to allow the work to be replicable?

Reviewer #1: Yes

4. Have the authors described where all data underlying the findings will be made available when the study is complete?

Reviewer #1: Yes

5. Is the manuscript presented in an intelligible fashion and written in standard English?

Reviewer #1: Yes

6. Review Comments to the Author

You may also provide optional suggestions and comments to authors that they might find helpful in planning their study.

Reviewer #1: Please note there is a linking word missing in the conclusion – line 389 - ‘is’ should be added as highlighted below:

The quality assessment of realist reviews 'is' essential

7. PLOS authors have the option to publish the peer review history of their article (what does this mean? ). If published, this will include your full peer review and any attached files.

**Do you want your identity to be public for this peer review?** For information about this choice, including consent withdrawal, please see our Privacy Policy .

Reviewer #1: **Yes: ** Tracey McConnell

---

## [Editor Report · Acceptance letter]

PONE-D-25-04980R1

PLOS ONE

Dear Dr. Mukumbang,

I'm pleased to inform you that your manuscript has been deemed suitable for publication in PLOS ONE. Congratulations! Your manuscript is now being handed over to our production team.

Kind regards,

on behalf of

Dr. Nishant Premnath Jaiswal

Academic Editor

PLOS ONE